# Impact of Adjuvant Radiotherapy in Patients with Central Neurocytoma: A Multicentric International Analysis

**DOI:** 10.3390/cancers13174308

**Published:** 2021-08-26

**Authors:** Laith Samhouri, Mohamed A. M. Meheissen, Ahmad K. H. Ibrahimi, Abdelatif Al-Mousa, Momen Zeineddin, Yasser Elkerm, Zeyad M. A. Hassanein, Abdelsalam Attia Ismail, Hazem Elmansy, Motasem M. Al-Hanaqta, Omar A. AL-Azzam, Amr Abdelaziz Elsaid, Christopher Kittel, Oliver Micke, Walter Stummer, Khaled Elsayad, Hans Theodor Eich

**Affiliations:** 1Department of Radiation Oncology, University Hospital Münster, Münster 48149, Germany; laithsamhouri@gmail.com (L.S.); Christopher.Kittel@ukmuenster.de (C.K.); hans.eich@ukmuenster.de (H.T.E.); 2Alexandria Clinical Oncology Department, Alexandria University, Alexandria 21500, Egypt; mohamed.meheissen@alexmed.edu.eg (M.A.M.M.); zeyadabdelaziz@hotmail.com (Z.M.A.H.); salam61@yahoo.com (A.A.I.); amrelsaid@yahoo.com (A.A.E.); 3Specialized Universal Network of Oncology (SUN), Alexandria 21500, Egypt; yelkerm@yahoo.com (Y.E.); hazemelmansy@hotmail.com (H.E.); 4Department of Radiotherapy and Radiation Oncology, King Hussein Cancer Center, Amman 11942, Jordan; aibrahimi@khcc.jo (A.K.H.I.); aalmousa@khcc.jo (A.A.-M.); 5Department of Pediatrics, King Hussein Cancer Center, Amman 11942, Jordan; MZ.12765@khcc.jo; 6Cancer Management and Research Department, Medical Research Institute, Alexandria University, Alexandria 21500, Egypt; 7Military Oncology Center, Royal Medical Services, Amman 11942, Jordan; Drhanaqta@yahoo.com; 8Princess Iman Research Center, King Hussein Medical Center, Royal Medical Services, Amman 11942, Jordan; Omarazzam89@outlook.com; 9Department of Radiotherapy and Radiation Oncology, Franziskus Hospital Bielefeld, 33699 Bielefeld, Germany; Oliver.Micke@franziskus.de; 10Department of Neurosurgery, University Hospital Münster, 48149 Münster, Germany; Walter.Stummer@ukmuenster.de

**Keywords:** neurocytoma, toxicities, rare tumors, management, radiation therapy

## Abstract

**Simple Summary:**

Central neurocytoma is a rare tumor accounting for <0.5% of all intracranial tumors. We analyzed 33 patients treated with surgical resection with or without radiotherapy from ten closely cooperating institutions in Germany, Egypt, and Jordan. Patients who received radiotherapy had longer progression-free survival with an acceptable toxicity profile.

**Abstract:**

Background: Central neurocytoma (CN) is a rare tumor accounting for <0.5% of all intracranial tumors. Surgery ± radiotherapy is the mainstay treatment. This international multicentric study aims to evaluate the outcomes of CNs patients after multimodal therapies and identify predictive factors. Patients and methods: We retrospectively identified 33 patients with CN treated between 2005 and 2019. Treatment characteristics and outcomes were assessed. Results: All patients with CN underwent surgical resection. Radiotherapy was delivered in 19 patients. The median radiation dose was 54 Gy (range, 50–60 Gy). The median follow-up time was 56 months. The 5-year OS and 5-year PFS were 90% and 76%, respectively. Patients who received radiotherapy had a significantly longer PFS than patients without RT (*p* = 0.004) and a trend towards longer OS. In addition, complete response after treatments was associated with longer PFS (*p* = 0.07). Conclusions: Using RT seems to be associated with longer survival rates with an acceptable toxicity profile.

## 1. Introduction

Central neurocytoma (CN) is a rare disease accounting for only ≤0.5% of all intracranial neoplasms originating from the ventricular space [1,2,3]. According to the recent World Health Organization (WHO) classification, CNs are classified as grade 2 and usually occur in young patients and adolescents, with a similar incidence between males and females [3,4]. Most CNs are well-differentiated and have a benign nature with favorable prognoses following the multimodal treatments [3,4]. However, malignant variants have been reported with an MIB-labeling index >2% with a higher recurrence rate [5,6,7]. Surgical resection is the mainstay of treatment of NC; however, residual or recurrent CNs are challenging to manage. The most important prognostic factor affecting patients’ outcomes is the extent of surgery [8,9]. The role of radiotherapy and chemotherapy remains controversial with a limited number of studies due to disease rarity.

This international multicenter study aims to evaluate the outcomes of CNs patients after multimodal therapies and identify other predictive factors which may influence the outcome.

## 2. Patients and Methods

Thirty-three patients with neurocytoma were collected between 2001 and 2019 from ten closely cooperating institutions in Germany, Egypt, and Jordan. Patient characteristics are summarized in Table 1. All patients with NC were presented in a multidisciplinary tumor board following surgery. After resection, almost all patients had received MRI (n = 32) and CT (n = 33) to define any residuals. The planning target volume (PTV) represented a 5–10 mm of the clinical target volume, an anatomically constrained 10–15 mm expansion of the gross-residual tumor and tumor bed.

From the 19 patients in RT cohort, 15 (79%) were treated with three-dimensional conformal RT (3D-CRT) and four (21%) with intensity-modulated radiation therapy (IMRT). The median cumulative RT dose was 54 Gy (range, 50–60 Gy), and it was delivered in 1.8–2 Gy daily fractions. All patients completed the radiation course without RT breaks. Patients were followed regularly every three months with MRI or CT scans to exclude tumor progression. Only two patients (6%) received chemotherapy. Common terminology criteria for adverse events (CTCAEs) has been used during and after RT to assess toxicities. Imaging data were reviewed for response assessment according to the recently updated RANO classification of malignant glioma. At the final analysis, two patients had died, while twenty-six were alive, with five patients lost to follow-up.

### Statistical Analysis

All statistical analyses were conducted with SPSS version 27.0 software (IBM, Armonk, NY, USA). Overall survival (OS) was calculated from the first day of RT and progression-free survival (PFS) was calculated from the TT until documented relapse or death. Time-dependent event curves were calculated using the Kaplan-Meier method and were compared using the log-rang test. Differences were considered statistically relevant at a *p*-value < 0.05.

## 3. Results

Radiotherapy (RT) has been applied in 19 patients (adjuvant RT in 18 patients and salvage RT in two patients). Fourteen patients were only operated without adjuvant therapy. The involved sites included the ventricles (43%), central (36%), and other locations (21%). At time of RT, ten patients had an Eastern Co-operative of Oncology Group (ECOG) score of 0, seven patients had a score of 1, and two patients had score of 2. Sex, tumor location, chemotherapy, RT dose, WHO grades, and *Ki67 MIB1 value* were equally distributed between RT and non-RT cohorts (*p* > 0.05). However, more patients in the RT group had residual tumors compared with the non-RT group (90% vs. 50%, *p* = 0.02). The median follow-up time was 56 months.

### 3.1. Overall and Progression-Free Survival Rates

The 5-year OS and 5-year PFS were 90 ± 7% and 76 ± 11%, respectively. Regarding OS, we could not observe any significant differences between the WHO grades (*p* = 0.8), site of lesion (*p* = 0.3), total resection (*p* = 0.4), chemotherapy administration (*p* = 0.7), intent of radiation (*p* = 0.4), radiation techniques (*p* = 1), and complete response after therapy (*p* = 0.2). However, patients who received RT had a trend towards longer OS than patients who did not (*p* = 0.09; Figure 1A). Concerning PFS, there were no significant differences between the WHO grades (*p* = 0.1), site of lesion (*p* = 0.3), total resection (*p* = 0.7), chemotherapy administration (*p* = 0.6), and radiation techniques (*p* = 1). On the other hand, patients who received RT had a significantly longer PFS than patients without RT (*p* = 0.004; Figure 1B), while complete response after therapy was associated with longer PFS (*p* = 0.07). Regarding radiation dose, no PFS or OS differences have been detected (*p* > 0.05).

### 3.2. Radiotherapy Toxicities

RT was well-tolerated without significant adverse events (AEs). However, half of the patients had toxicities (21% with grade 1 and 32% with grade 2 AEs). The most common toxicities were partial alopecia, fatigue, and skin redness. Local grade 1 or 2 alopecia and skin changes were described as late RT toxicity in six patients. No patients had grade 3 or 4 AEs. There were no grade 3 or 4 chronic toxicities.

## 4. Discussion

This is an international multicentric analysis that investigated the role of radiotherapy in neurocytoma. Surgery is the standard treatment for NC patients; complete resection is infeasible in locally advanced cancer [9]. RT has a significant effect on PFS and OS, even for tumors with high-risk features, as previously reported by various authors [8].

Several studies, including systematic reviews and meta-analyses, demonstrate a rational indication for RT in recurrent or residual tumors [10,11,12]. In our study, we focused on the role of fractionated radiotherapy (FCRT), however several studies, including systemic reviews, investigated and compared the results between FCRT and stereotactic radiosurgery (SRS) [13,14,15,16,17,18,19,20,21,22,23]. Therefore, we should balance the benefits and risks of RT. In addition, we should consider that both techniques have over 80–90% long-term local control rates [14]. However, some of these studies preferred SRS owing to lower toxicities rates and the relative risks of local recurrence. In addition, SRS might reduce the inconvenience and delayed toxicity of FCRT due to its higher conformality and smaller target volume [14]. A literature summary table presenting different radiotherapy techniques for neurocytoma patients is provided (Table 2).

The PFS rate in our study is excellent, even in patients who underwent subtotal resection and adjuvant RT or recurrent RT without surgery. Our study reports a five-year PFS of 76% and OS reaching 90%, consistent with the previous reports. Tumor location seems to be irrelevant to PFS and OS improvement. In subgroup analysis, we found no significant difference between WHO grades, lesion site, total resection status, chemotherapy administration, the intent of radiation, and radiation techniques. RT administration correlated significantly with PFS (*p* = 0.004), while complete response after treatment seems to be associated with better PFS (*p* = 0.07). In terms of OS, patients who received radiotherapy had a trend towards longer OS than patients without RT (*p* = 0.09). However, 90% of irradiated patients underwent subtotal resection.

The optimal radiation dose for CN patients was investigated by Rades et al. [9]. In our analysis, most of the patients received a total dose of 54 Gy and higher doses (>54 Gy) were not associated with better clinical outcomes (*p* = 0.05). Thus, a cumulative dose of 54 Gy may be appropriate for CN patients regardless of the resection status.

In our study, RT was well-tolerated without significant adverse events. No patients had grade 3 or 4 AEs, in contrast to several studies which showed higher neurological toxicities following treatment [10,27,28,29,30,31]. However, we should also consider the role of surgical resection in developing the side effects, especially late neurotoxicity [32,33,34]. In the recent EORTC trial, Klein and colleagues [35] proved that memory functioning was not associated with RT target volumes in low-grade glioma patients. Moreover, radiotherapy does not have a deleterious effect on memory function after one year of treatment, compared to chemotherapy.

The role of the MIB-1 labeling index may correlate with prognosis [8,36]. Unfortunately, we could not detect any significant impact of MIB rate on survivals, probably due to the small sample size.

Our study was limited with its retrospective design and limited number, although patient characteristics were reasonably distributed between the examined groups. Moreover, our data agree with previous reports and add to existing literature regarding the importance of RT in this rare disease entity. That being said, a need for international collaboration to create a prospective register is needed to confirm these results further.

## 5. Conclusions

Postoperative RT may to be associated with longer survival rates with an acceptable toxicity profile. However, a larger international prospective analysis is necessary to prove the role of RT.

## Figures and Tables

**Figure 1 cancers-13-04308-f001:**
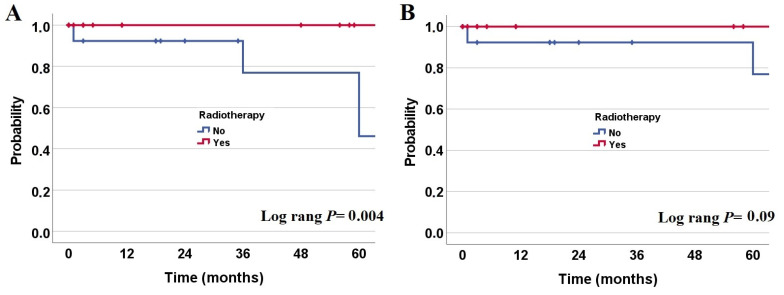
Kaplan–Meier estimate of progression-free survival (**A**) and overall survival (**B**) in central neurocytoma patients according to radiotherapy administration (N = 33).

**Table 1 cancers-13-04308-t001:** Treatment characteristics and postoperative therapy.

Characteristic	Nr. (% or Range)	Therapy		
		Radiotherapy	No Radiotherapy	*p*-Value
Patients	33	19 (58%)	14 (42%)	
Med. age (range)	25 y (4–58)	24 (12–58)	26 (4–50)	0.5
Sex	M: 17 (51%) F: 16 (49%)	9 (47%) 10 (53%)	8 (47%) 6 (53%)	0.7
Ki67 MIB1 value, median	8 (1–30)	7.5 (1–30)	10 (1–25)	0.8
Resection				0.02
Gross total resection	9 (27%)	2 (10%)	7 (50%)	
Subtotal resection	24 (73%)	17 (90%)	7 (50%)	
Chemotherapy				0.2
Yes	2 (6%)	0 (0%)	2 (14%)	
No	31 (94%)	19 (100%)	12 (86%)	
WHO grade				0.6
I	5 (15%)	2 (10%)	3 (21%)	
II	25 (76%)	15 (80%)	10 (72%)	
III	1 (3%)	1 (5%)	0 (0%)	
Unknown	2 (6%)	1 (5%)	1 (7%)	
Primary tumor site				0.7
Ventricles	14 (42%)	7 (37%)	7 (50%)	
Central	12 (36%)	7 (37%)	5 (36%)	
Others	7 (21%)	5 (26%)	2 (14%)	
Relapse pattern				0.4
Yes	7 (21%)	3 (16%)	4 (29%)	
No	26 (79%)	16 (84%)	10 (71%)	

M, males; F, females.

**Table 2 cancers-13-04308-t002:** Review of literature.

Studies	Number of Patients Receiving RT/All Patients	Median RT Dose in Gy (Range)	Local Control Rate
**Fractionated Conventional Radiotherapy (FCRT)**			
Sharma et al. 1998 [24]	15/15	40–60	100%
Rades et al. 2006 [9]	177/350	50–60	87%
Leenstra et al. 2007 [11]	18/18	Median: 54.5 (48.6–61.2)	78%
Kim et al. 2013 [15]	7/58	Median: 54 (50.4–55.8)	80%
Chen et al. 2014 [25]	63/63	Median: 54 (46–60)	100%
Byun et al. 2018 [26]	10/40	54–56	69%
Current study	19/33	Median: 54 (50–60)	84%
**Stereotactic Radiosurgery (SRS)**			
Yen et al. 2007 [13]	6/6	Median: 15.1 (9–20)	100%
Kim et al. 2007 [16]	7/13	Median: 15.7 (15–18)	85%
Matsunaga et al. 2010 [17]	7/7	Median: 13.9 (12–18)	88%
Genc et al. 2011 [18]	18/18	Median: 16.7 (9–20)	93%
Karlsson et al. 2012 [19]	35/35	Median: 14.0 (11–25)	83%
Kim et al. 2013 [15]	17/58	Median: 16 (9–20)	80%
Monaco at al. 2015 [14]	8/8	Median: 14.6 Gy (12–20)	87%

RT: radiotherapy; SRS: stereotactic radiosurgery; FCRT: fractionated conventional radiotherapy.

## Data Availability

The data presented in this study are available on request from the corresponding author. The data are not publicly available due to privacy.

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
