# Peer review of "Impact of Adjuvant Radiotherapy in Patients with Central Neurocytoma: A Multicentric International Analysis"

_cancers, 2021, doi:10.3390/cancers13174308_

Round 1

Reviewer 1 Report

The authors have submitted a manuscript on adjuvant radiotherapy in patients with central neurocytoma. The data were based on a multicentric international analysis. The manuscript is excellently written, concise and the results are sound, although based on retrospective data. The manuscript should be considered for publication provided that the following aspects are implemented.

Discussion, second paragraph

The authors state several studies including systematic reviews and include three references dating back between 2006 and 2011. These data should be updated and discussed.

Igor J Barani , David R Raleigh , David Larson The management of central neurocytoma: radiotherapy. Neurosurg Clin N Am . 2015 Jan;26(1):45-56

Roxanna M Garcia, Michael E Ivan, Taemin Oh, Igor Barani, Andrew T Parsa. Intraventricular neurocytomas: a systematic review of stereotactic radiosurgery and fractionated conventional radiotherapy for residual or recurrent tumors Clin Neurol Neurosurg . 2014 Feb;117:55-64. doi: 10.1016/j.clineuro.2013.11.028. Epub 2013 Dec 7.

Anil K Mahavadi, Priyen M Patel, Manish Kuchakulla, Ashish H Shah, Dan Eichberg, Evan M Luther, Ricardo J Komotar, Michael E Ivan.Central Neurocytoma Treatment Modalities: A Systematic Review Assessing the Outcomes of Combined Maximal Safe Resection and Radiotherapy with Gross Total Resection. World Neurosurg . 2020 May;137:e176-e182. doi: 10.1016/j.wneu.2020.01.114. Epub 2020 Jan 27.

G Virbel, H Cebula, A Coca, B Lhermitte, L Bauchet, G Noël [Choice optimisation of radiation therapy technique for central neurocytomas from literature data] [Article in French] Cancer Radiother . 2020 Dec;24(8):882-891. doi: 10.1016/j.canrad.2020.03.011. Epub 2020 Aug 1.

Additionally the role of radiosurgery should be discussed in more detail with respect to fractionated approaches. In particular the authors should address the findings from Garcia et al. The authors also address the risk for late effects in particular neurocognitive dysfunction and state articles also dating back to 2002 and 2007. The recent findings of the EORTC trial 22033/26033 should be added and discussed.

Klein M, Drijver AJ, van den Bent MJ, Bromberg JC, Hoang-Xuan K, Taphoorn MJB, Reijneveld JC, Ben Hassel M, Vauleon E, Eekers DBP, Tzuk-Shina T, Lucas A, Freixa SV, Golfinopoulos V, Gorlia T, Hottinger AF, Stupp R, Baumert BG. Memory in low-grade glioma patients treated with radiotherapy or temozolomide: a correlative analysis of EORTC study 22033-26033. Neuro Oncol. 2021 May 5;23(5):803-811

Neurocytoma is a rare disease. Numerous retrospective were published in the past. A table illustrating these findings including the data that were submitted for this journal would help the reader to understand the current limitations of the role of radiotherapy. This table should also include data on radiosurgery.

Author Response

The authors state several studies including systematic reviews and include three references dating back between 2006 and 2011. These data should be updated and discussed: Igor J Barani et al. 2015, Roxanna M Garcia et al. 2014, Anil K Mahavadi et al. 2020, G Virbel et al. 2020

  • We followed your advice and added all recommended references to the discussion section (page 5). In addition, we added ‘’table 2’’ to provide the reader with an overview of outcome of different radiation techniques.

Additionally the role of radiosurgery should be discussed in more detail with respect to fractionated approaches. In particular the authors should address the findings from Garcia et al. The authors also address the risk for late effects in particular neurocognitive dysfunction and state articles also dating back to 2002 and 2007. The recent findings of the EORTC trial 22033/26033 should be added and discussed: Klein M et al. 2021

  • This is an interesting suggestion; therefore, we discussed this point in details.
  • Recent EORTC trial regarding neurocognitive dysfunction following cerebral radiation has been added on page 6.

‘’However, we should also consider the role of surgical resection in developing the side effects, especially late neurotoxicity [29–31]. In the recent EORTC trial, Klein and colleagues [32] proved that memory functioning was not associated with RT target volumes in low-grade glioma patients. Moreover, radiotherapy does not have a deleterious effect on memory function after one year of treatment compared to chemotherapy. ‘’

Neurocytoma is a rare disease. Numerous retrospective were published in the past. A table illustrating these findings including the data that were submitted for this journal would help the reader to understand the current limitations of the role of radiotherapy. This table should also include data on radiosurgery.

  • A literature summary table provides differenent radiotherapy techniques for neurocytoma patients has been added tot he manuscript (table 2).

Reviewer 2 Report

The authors present results of an international study focused on a value of adding radiotherapy to surgery in the treatment of neurocytoma.

The value of the study comes from a relatively high sample size of the rare neoplasm which still needs recommendations for treatment.

The study confirms the expected value of irradiation. The authors did not find the extend of surgery a significant factor for survival rates however it should be emphasised that 90% of irradiated patients underwent subtotal resection.

The authors did not define the "complete response" criteria which would be a valuable detail especially in a multicentric study. I recommend to supplement the text with this data. 

I also recommend to summarize long term toxicities of the treatment.

Author Response

The authors present results of an international study focused on a value of adding radiotherapy to surgery in the treatment of neurocytoma. The value of the study comes from a relatively high sample size of the rare neoplasm which still needs recommendations for treatment. The study confirms the expected value of irradiation. The authors did not find the extend of surgery a significant factor for survival rates however it should be emphasised that 90% of irradiated patients underwent subtotal resection.

  • This is an important comment. Therefore, we adjusted the discussion and included the recommended sentence on page 7.
  • ‘‘ In terms of OS, patients who received radiotherapy had a trend towards longer OS than patients without RT (P=.09). However, 90% of irradiated patients underwent subtotal resection.’’

The authors did not define the "complete response" criteria which would be a valuable detail especially in a multicentric study. I recommend to supplement the text with this data. 

  • Imaging data were reviewed for response assessment according to the recently updated RANO classification of malignant glioma (Wen et al. J Clin Oncol . 2010 Apr 10;28(11):1963-72. doi: 10.1200/JCO.2009.26.354).

I also recommend to summarize long term toxicities of the treatment.

  • We followed your advice and added late toxicities tot he manuscript.

‘‘Local grade 1 or 2 alopecia and skin changes were described as late RT toxicity in six patients. No patients had grade 3 or 4 AEs. There were no grade 3 or 4 chronic toxicities.‘‘

Round 2

Reviewer 1 Report

Recommendations were considered. Manuscript can be accepted for publication.